

# Development of a predictive model for severe hyperlipidemic acute pancreatitis based on LASSO regression: a retrospective study

Qingyu Zhang[1], Runping Han[1,2], Zhenxing Li[1], Kunfeng Yan[1], Xiaorong Dai[1] and Gongchao Yu[1]

[1] Department of Gastroenterology, Taixing People's Hospital Affiliated to Yangzhou University, Taizhou, China
[2] Department of Clinical Medicine, Jiangsu Health Vocational College, Nanjing, China

Corresponding author
Gongchao Yu,
bzyxyyugongchao@126.com

## ABSTRACT

**Background**. In recent years, the incidence of hyperlipidemic acute pancreatitis (HLAP) has been increasing. Identifying the risk factors associated with severe HLAP and developing a predictive model are crucial for early detection and intervention, thereby alleviating the disease burden. This study aimed to investigate the risk factors associated with severe HLAP and develop a predictive model.

**Methods**. Data on HLAP treated in Taixing People's Hospital Affiliated to Yangzhou University from January 1, 2020, to June 30, 2023, were retrospectively collected and divided into a mild group ($N = 296$) and a moderate severe/severe group ($N = 60$). Univariate analysis and Least Absolute Shrinkage and Selection Operator (LASSO) regression were used to select variables, and the selected variables were incorporated into logistic regression to analyze the risk factors of severe disease. A logistic regression model was constructed. The receiver operating characteristic (ROC) curve and the area under the curve (AUC) were used to evaluate model differentiation, and the Hosmer–Lemeshow goodness-of-fit test and calibration curve were used to evaluate model consistency.

**Results**. The univariate analysis showed statistically significant differences in 50 variables between the mild and moderately severe/severe groups. LASSO regression identified the following variables: D-dimer, blood calcium, cholesterol, standard bicarbonate (SB), total carbon dioxide, and C-reactive protein–albumin ratio (CAR). The constructed logistic regression model included D-dimer, blood calcium, and cholesterol, with an AUC of 0.8341 (95% CI [0.7724–0.8958]). The model's calibration was assessed using the Hosmer–Lemeshow goodness-of-fit test ($\chi^2 = 6.8383$, $P = 0.5542$), and the calibration curve demonstrated that the model's predictions closely aligned with observed outcomes.

**Conclusion**. The risk factors of severe HLAP include D-dimer elevation, calcium depletion and cholesterol elevation. The predictive model established by logistic regression has good performance, which is helpful for early identification and intervention by clinicians.

## INTRODUCTION

Hyperlipidemic acute pancreatitis (HLAP), defined as acute pancreatitis resulting from hyperlipidemia, is characterized by significantly increased triglyceride levels, moderately elevated amylase levels, pseudohyponatremia, and a tendency to occur at a younger age (*Expert Consensus on the Diagnosis and Treatment of Acute Pancreatitis with Hypertriglyceridemia (Emergency Department) (Expert Group), 2021*; *Zhou et al., 2024*; *Fan, Li & Wu, 2025*). In recent years, the incidence of HLAP has been increasing, making it the second most common cause of acute pancreatitis (*Chinese Pancreatic Surgery Association et al., 2021*). HLAP is associated with high morbidity and mortality rates, imposing a significant medical burden (*Deng et al., 2025*). Therefore, it is of great significance to study the risk factors for HLAP-related severe cases and develop a prediction model, in order to identify severe cases early and initiate early and active intervention, thereby reducing the incidence of severe cases, mortality rate, and alleviating the disease burden.

A number of scholars have studied the risk factors related to the severe progression of HLAP, such as serum calcium, D-dimer, neutrophil percentage, C-reactive protein (CRP), and blood urea nitrogen (BUN). The research methods have mostly included single-factor analysis and logistic regression analysis (*Thong, Mong Trinh & Phat, 2021*). Some scholars have studied the predictive performance of a single indicator or combined indicators for severe HLAP (*Zhu et al., 2020*), while others studied the predictive performance of commonly used acute pancreatitis scoring systems for severe HLAP (*Qiu et al., 2015*; *Yang et al., 2016*). However, there are few studies that specifically target the construction of a model for predicting severe HLAP (*Dong et al., 2024*; *Lin et al., 2024*). No prediction models or scoring systems have been recommended for clinical use by guidelines or consensus statements so far. Therefore, developing early diagnostic prediction models with excellent performance that are specifically designed for the severe progression of HLAP in the early stage is urgently needed. LASSO regression is an effective variable selection tool that can solve the problems of collinearity and overfitting, which is particularly important in the era of big data (*Emmert-Streib & Dehmer, 2019*). In this context, in this study, we collected case data of HLAP patients treated at Taixing People's Hospital Affiliated to Yangzhou University, used LASSO regression to screen and analyze risk factors, and constructed a logistic model to help clinicians accurately identify severe patients in the early stage.

## MATERIALS & METHODS

### Study population

The clinical data of patients with HLAP admitted to Taixing People's Hospital Affiliated to Yangzhou University from January 1, 2020, to June 30, 2023, were collected. Inclusion criteria: the patients were included if they met the diagnostic criteria of HLAP (referring to the guideline "Expert Consensus on the Diagnosis and Treatment of Emergency Hypertriglyceridemic Acute Pancreatitis": (1) meeting the diagnostic criteria of acute pancreatitis (AP); (2) serum TG level ≥1,000 mg/dL (11.30 mmol/L), or serum TG level 500–1,000 mg/dL (5.65–11.30 mmol/L) with chylous serum; and other causes of AP excluded). Exclusion criteria: (1) severe liver and kidney function diseases; (2) malignant

tumors; (3) transfer to another hospital halfway; (4) the interval between onset and admission greater than 48 h; and (5) incomplete data (missing >20%).

## Data collection and sorting

Data were entered using Epidata3.1, with double entry and consistency checks conducted. The data collected included gender, age, vital signs, history of diabetes (type 1 or type 2), recurrence, blood routine, liver and kidney function, electrolytes, coagulation function, blood gas analysis, blood glucose, blood lipid, amylase and other data within 48 h after admission. Excel and R 4.3.1 were used to organize the data. The makeX function in glmnet package of R 4.3.1 software was used to complete the missing values by mean imputation when LASSO regression was performed, and the simple median imputation method was used to handle the missing values when the logistic regression model was constructed.

## Risk factor analysis and model construction

The patients were classified into the mild group and the moderate/severe group based on the 2012 American Atlanta Acute Pancreatitis New Grading and Classification System. First, univariate analysis was conducted. Continuous variables were tested for normality using the Shapiro–Wilk test, and were described using the mean ± standard deviation if they met the normal distribution criteria. In that case, the t test was used. If they did not meet the normal distribution criteria, they were described using the median (interquartile range (IQR)), and the rank-sum test was used. Categorical variables were described using the number of cases, and the chi-square test or Fisher's exact test was used. $P < 0.05$ was considered statistically significant. The statistically significant variables identified in the univariate analysis were included in the LASSO regression analysis using the glmnet package of R 4.3.1 software, with a 10-fold cross-validation method, and the lambda value (lambda.1se) corresponding to the minimum mean squared error was selected as the optimal solution. The variables selected by LASSO regression were used to construct the logistic regression model using the rms package in R 4.3.1 software. The model's discriminatory ability was evaluated using the ROC curve and the AUC. The consistency of the model was evaluated using the Hosmer–Lemeshow goodness-of-fit test and calibration curve.

## Ethics approval

This study was approved by the Medical Ethics Committee of Taixing People's Hospital (No: LS2025008; No: YJ2025009), and the requirement for informed consent was waived.

# RESULTS

## Patients' demographic characteristics and univariate analysis

A total of 356 patients were enrolled in this study, including 237 males (66.6%) and 119 females (33.4%), aged 42.7 ± 10.2 years. There were 296 mild cases (83.1%) and 60 moderate/severe cases (16.8%). The mean length of hospital stay was 7.90 days, with a median of 7 days and an IQR of 5–9 days. A total of 243 patients (68.3%) had fatty liver, 147 (41.3%) had a history of diabetes (type 1 or type 2), and 111 (31.2%) had recurrent

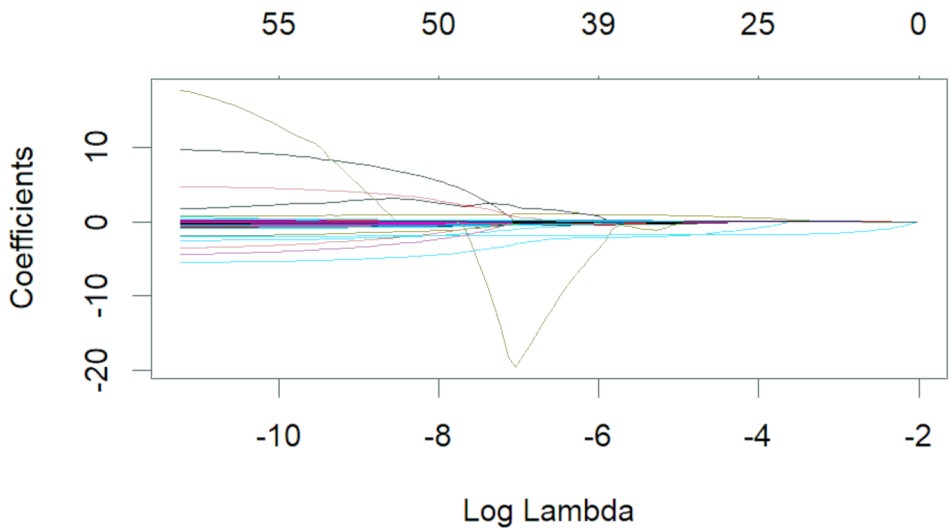

**Figure 1** Lambda and partial regression coefficent of LASSO regression.

HLAP. A total of 107 indicators (variables) were collected. Univariate analysis identified 50 variables with statistically significant differences between the mild and moderate/severe groups (Table 1).

## Variables selection of LASSO regression

The 50 variables identified through the univariate analysis were further analyzed using 10-fold cross-validation LASSO regression (Figs. 1 and 2), with Lambda.1se = 0.076 selected as the optimal solution. At this time, the screened variables were D-dimer, serum calcium, cholesterol, SB, total carbon dioxide, and CAR (Table 2).

## Logistic regression model

The abovementioned variables were included in the multivariate logistic regression analysis, and the results showed that D-dimer, serum calcium, and cholesterol were the risk factors for severe HLAP (AUC = 0.8341 (95% CI [0.7724–0.8958]); Table 3). The ROC curve graph is presented in Fig. 3, showing that the model has good discriminatory ability. The Hosmer–Lemeshow goodness-of-fit test was conducted ($\chi^2$ = 6.8383, $P$ = 0.5542). The calibration curve graph is shown in Fig. 4, where the curve is close to the ideal curve, indicating that the model is consistent.

## DISCUSSION

This study utilized LASSO regression for variable selection and logistic regression modeling, identifying elevated D-dimer levels, decreased serum calcium, and increased cholesterol as significant risk factors for severe HLAP. These findings are consistent with many previous studies (*Zhou et al., 2024*; *Zhu et al., 2020*; *Dong et al., 2024*; *Lin et al., 2024*; *Wu et al., 2025*; *Liu et al., 2023*; *Xu et al., 2022*). Current research suggests that the pathogenesis of HLAP primarily involves free fatty acids, impaired pancreatic microcirculation, inflammatory

**Table 1  Univariate analysis of severe hyperlipidemic acute pancreatitis.**

| Variable | Mild | Moderate/severe | $t/W/\chi^2$ | $P$ |
|---|---|---|---|---|
| Diabetes [$n$ (%)] | | | 6.1924 | 0.013 |
| No | 182 (51.27%) | 26(7.32%) | | |
| Yes | 113 (31.83%) | 34(9.58%) | | |
| Recurrence [$n$ (%)] | | | 6.2933 | 0.012 |
| No | 195 (54.78%) | 50 (14.04%) | | |
| Yes | 101 (28.37%) | 10 (2.81%) | | |
| Urinary protein [$n$ (%)] | | | | |
| No | 96 (48%) | 13 (6.5%) | 10.716 | 0.001 |
| Yes | 62 (31%) | 29 (14.5%) | | |
| Red blood cell count [$M$ ($P_{25}$, $P_{75}$), $10^{12}$ /L] | 4.80 (4.40, 5.14) | 5.02 (4.64, 5.44) | 6,816.5 | 0.022 |
| Hematocrit [$\bar{x} \pm s$, %] | 41.47 ± 4.66 | 43.45 ± 6.07 | −2.3549 | 0.021 |
| Red blood cell distribution width [$M$ ($P_{25}$, $P_{75}$), %] | 13.00 (12.40, 13.50) | 13.20 (12.90, 14.20) | 5,861 | 0.002 |
| White blood cell count [$M$ ($P_{25}$, $P_{75}$), $10^9$ /L] | 12.58 (10.09, 15.58) | 13.65 (11.60, 16.20) | 7,154 | 0.030 |
| Neutrophilic granulocyte percentage [$M$ ($P_{25}$, $P_{75}$), %] | 82.20 (76.82, 86.85) | 85.10 (80.95, 87.80) | 6,884.5 | 0.015 |
| Lymphocyte percentage [$M$ ($P_{25}$, $P_{75}$), %] | 10.70 (7.10, 15.60) | 7.90 (5.21, 11.55) | 10,102 | 0.002 |
| Eosinophils percentage [$M$ ($P_{25}$, $P_{75}$), %] | 0.40 (0.10, 0.80) | 0.10 (0, 0.50) | 10,366 | <0.001 |
| Lymphocyte count [$M$ ($P_{25}$, $P_{75}$), $10^9$/L] | 1.32 (0.94, 1.82) | 1.06 (0.79, 1.60) | 9,577 | 0.015 |
| Eosinophil count [$M$ ($P_{25}$, $P_{75}$), $10^9$/L] | 0.04 (0.02, 0.09) | 0.02 (0, 0.06) | 10,272 | <0.001 |
| Platecrit [$M$ ($P_{25}$, $P_{75}$), %] | 0.23 (0.19, 0.27) | 0.26 (0.20, 0.30) | 5,325.5 | 0.014 |
| C-reactive protein [$M$ ($P_{25}$, $P_{75}$), mg/L] | 30.50 (7.48, 101.00) | 90.7 (21.20, 253.94) | 3,220 | <0.001 |
| Procalcitonin [$M$ ($P_{25}$, $P_{75}$), ng/mL] | 0.07 (0.04, 0.17) | 0.33 (0.06, 1.55) | 3,490 | <0.001 |
| Prothrombin time, PT [$M$ ($P_{25}$, $P_{75}$), s] | 11.30 (10.70, 11.85) | 11.80 (11.30, 12.50) | 3,223.5 | <0.001 |
| Prothrombin activity, PTA [$\bar{x} \pm s$, %] | 105.69 ± 15.11 | 94.65 ± 15.69 | 4.264 | <0.001 |
| International normalized ratio, INR [$M$ ($P_{25}$, $P_{75}$)] | 0.98 (0.93, 1.03) | 1.02 (0.98, 1.10) | 3,355 | <0.001 |
| Thrombin time, TT [$M$ ($P_{25}$, $P_{75}$), s] | 12.90 (12.20, 13.75) | 11.90 (11.55, 12.60) | 1,122 | 0.015 |
| D-dimer [$M$ ($P_{25}$, $P_{75}$), ng/ml] | 550 (343, 884) | 1,415 (726, 2,070) | 1,392.5 | <0.001 |
| Na [$M$ ($P_{25}$, $P_{75}$), mmol/L] | 133.10 (129.00, 136.60) | 131.55(125.25, 135.05) | 10,221 | 0.011 |
| Ca [$M$ ($P_{25}$, $P_{75}$), mmol/L] | 2.26 (2.14, 2.39) | 2.03(1.77, 2.26) | 11,922 | <0.001 |
| Phosphorus [$M$ ($P_{25}$, $P_{75}$), mmol/L] | 1.12(0.90, 1.39) | 1.02 (0.71, 1.31) | 9,782.5 | 0.012 |
| $HCO_3^-$ [$M$ ($P_{25}$, $P_{75}$), mmol/L] | 20.50 (17.85, 22.10) | 18.10(13.10, 20.80) | 8,144.5 | <0.001 |
| Urea nitrogen [$M$ ($P_{25}$, $P_{75}$), mmol/L] | 4.58 (3.68, 5.65) | 5.05(4.30, 7.20) | 3,886 | 0.032 |
| Uric acid [$M$ ($P_{25}$, $P_{75}$), umol/L] | 352 (279, 433) | 401 (310, 504) | 5,701 | 0.005 |
| AST [$M$ ($P_{25}$, $P_{75}$), U/L] | 23.00 (18.50, 32.00) | 29.50 (22.50, 39.50) | 1,399.5 | 0.047 |
| LDH [$M$ ($P_{25}$, $P_{75}$), U/L] | 246 (205, 325) | 341 (271, 488) | 3,153 | <0.001 |
| Total cholesterol, TC [$M$ ($P_{25}$, $P_{75}$), mmol/L] | 10.06 (7.96, 13.29) | 14.50 (9.92, 18.69) | 5,267.5 | <0.001 |
| Triglyceride, TG [$M$ ($P_{25}$, $P_{75}$), mmol/L] | 19.39(15.03, 25.85) | 25.07(17.66, 28.57) | 6,294.5 | <0.001 |
| Low density lipoprotein, LDL [$M$ ($P_{25}$, $P_{75}$), mmol/L] | 2.21 (1.19, 3.14) | 2.71 (1.36, 4.10) | 5,863 | 0.037 |
| Apolipoprotein B [$M$ ($P_{25}$, $P_{75}$), g/L] | 0.53 (0.31, 0.82) | 0.68 (0.37, 1.37) | 4,201.5 | 0.023 |
| Apolipoprotein A1/B [$M$ ($P_{25}$, $P_{75}$)] | 1.82 (1.10, 3.10) | 1.18 (0.61, 2.20) | 7,027 | <0.001 |
| CK-MB [$M$ ($P_{25}$, $P_{75}$), U/L] | 2.88(1.09, 8.60) | 8.00 (1.56, 11.00) | 3,468.5 | 0.005 |
| Serum amylase [$M$ ($P_{25}$, $P_{75}$), IU/L] | 162(96, 292) | 403 (215, 740) | 3,110 | <0.001 |

 

| Variable | Mild | Moderate/severe | $t/W/\chi^2$ | $P$ |
|---|---|---|---|---|
| PH [$M$ ($P_{25}$, $P_{75}$)] | 7.39 (7.36, 7.42) | 7.34 (7.26, 7.40) | 2,797 | <0.001 |
| pCO$_2$ [$M$ ($P_{25}$, $P_{75}$), mmHg] | 40(37,42) | 35 (29, 39) | 2,788 | <0.001 |
| actual bicarbonate, AB [$M$ ($P_{25}$, $P_{75}$), mmol/L] | 24.70 (21.35, 26.00) | 19.55 (12.70, 25.80) | 2,664 | <0.001 |
| Standard bicarbonate, SB [$M$ ($P_{25}$, $P_{75}$), mmol/L] | 24.75 (22.45, 25.80) | 21.25 (14.20, 24.7) | 2,702 | <0.001 |
| Total CO$_2$ [$M$ ($P_{25}$, $P_{75}$), mmol/L] | 26 (22.10, 27) | 20.50 (14, 27) | 2,660.5 | <0.001 |
| Standard base excess, SBE [$M$ ($P_{25}$, $P_{75}$), mmol/L] | −0.20 (−3.80, 1.40) | −5.90 (−15, 0.70) | 2,684.5 | <0.001 |
| Actual base excess, ABE [$M$ ($P_{25}$, $P_{75}$), mmol/L] | −0.20 (−3.20, 1.20) | −4.70 (−13.60, −0.30) | 2,713 | <0.001 |
| Lactic acid [$M$ ($P_{25}$, $P_{75}$), mmol/L] | 1.70 (1.30, 2.40) | 2.30 (1.55, 2.95) | 1,447.5 | 0.006 |
| Ionized calcium [$M$ ($P_{25}$, $P_{75}$), mmol/L] | 1.00 (0.94, 1.06) | 0.95 (0.79, 1.03) | 2,536 | 0.012 |
| Blood glucose [$M$ ($P_{25}$, $P_{75}$), mmol/L] | 9.97 (7.34, 14.14) | 14.78 (10.55, 17.66) | 4,275 | <0.001 |
| NLR [$M$ ($P_{25}$, $P_{75}$)] | 7.61 (4.93, 12.09) | 11.00 (7.33, 17.01) | 5,896.5 | 0.002 |
| PLR [$M$ ($P_{25}$, $P_{75}$)] | 145.70 (102.26, 215.04) | 210.77 (124.87, 290.08) | 6,085.5 | 0.005 |
| MLR [$M$ ($P_{25}$, $P_{75}$)] | 0.51 (0.35, 0.77) | 0.69 (0.32, 1.26) | 6,563 | 0.034 |
| CLR [$M$ ($P_{25}$, $P_{75}$)] | 24.86 (4.89, 85.92) | 92.41 (22.04, 227.91) | 2,809 | <0.001 |
| CAR [$M$ ($P_{25}$, $P_{75}$)] | 0.73 (0.17, 2.31) | 3.28 (0.43, 6.64) | 2,620.5 | <0.001 |

**Notes.**

NLR, neutrophil/lymphocyte; PLR, platelet/lymphocyte; MLR, monocytes/lymphocytes; CLR, C-reactive protein/lymphocyte; CAR, C-reactive protein/albumin.

The $t$-test was used for continuous variables with normal distribution. The wilcoxon rank sum test was used for continuous variables with non-normal distribution. The chi-square test or Fisher's exact test was used for categorical variables. $P < 0.05$ was considered statistically significant.

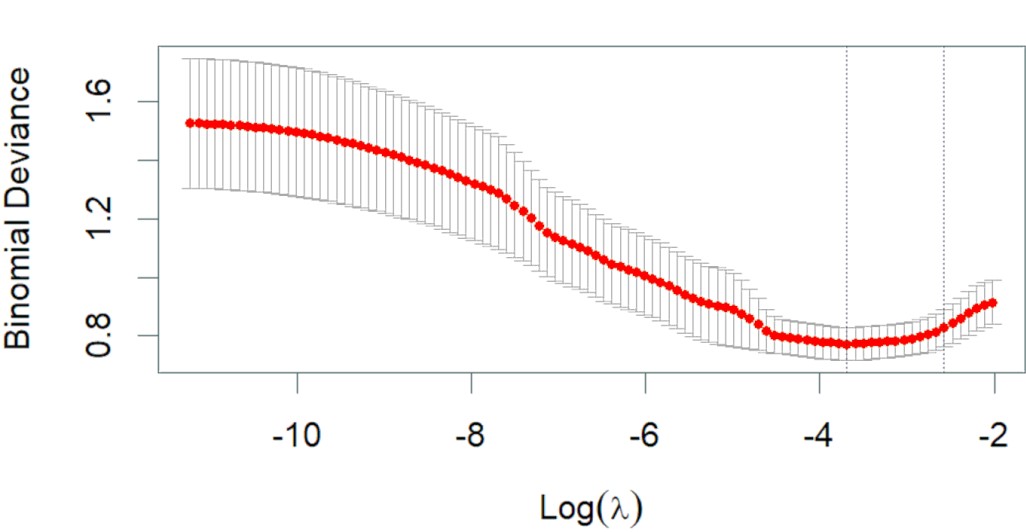

**Figure 2** Lambda and deviation plot of LASSO regression analysis.

response, oxidative stress, and intrapancreatic calcium overload (*Qiu et al., 2023*). D-dimer is a degradation product of fibrin, and an elevated D-dimer level indicates the presence of a state of hypercoagulability and secondary fibrinolysis activation. This study identified D-dimer as a risk factor for severe HLAP, which may be because HLAP releases large amounts of inflammatory mediators, activates the coagulation and fibrinolysis

**Table 2  Partial regression coefficients of non-return-to-zero variable of lasso regression (Lambda= Lambda.1se).**

| Variable | Partial regression coefficients |
|---|---|
| D-dimer | 0.0002 |
| Ca | −1.2003 |
| Total cholesterol, TC | 0.0409 |
| Standard bicarbonate, SB | −0.0131 |
| Total $CO_2$ | −0.0275 |
| C-reactive protein/albumin | 0.0037 |

**Table 3  Logistic regression model.**

| | Coefficients | Standard error | Wald Z | P | OR | 95%CI |
|---|---|---|---|---|---|---|
| Constant | 2.7796 | 1.5088 | 1.84 | 0.065 | 16.1118 | 0.8372–310.0716 |
| D-dimer | 0.0009 | 0.0003 | 3.59 | <0.001 | 1.0009 | 1.0004–1.0015 |
| Ca | −3.4379 | 0.6803 | −5.05 | <0.001 | 0.0321 | 0.0085–0.1219 |
| Total cholesterol, TC | 0.1744 | 0.0326 | 5.35 | <0.001 | 1.1905 | 1.1168–1.2691 |

**Notes.**

The significance of coefficients was assessed using the Wald test. $P < 0.05$ was considered statistically significant.

system, forms microvascular thrombi, and impairs pancreatic microcirculation, thereby exacerbating the condition (*Dong et al., 2024*; *Wu et al., 2025*). Previous studies have shown that hypocalcemia often indicates a severe condition, which may be related to the binding of fatty acids to calcium ions and intrapancreatic calcium overload (*Lin et al., 2024*). Therefore, monitoring serum calcium is of great significance. This study also revealed cholesterol's predictive role for severe HLAP, which is consistent with the research of *Liu et al. (2023)*, and may be related to the mechanisms underlying intrapancreatic calcium overload and directly damaging endothelial cells in pancreatic cells (*Xu et al., 2022*). Many previous studies have shown that the ratio of neutrophils to lymphocytes, CRP, PCT, blood glucose, and BUN are risk factors. Our univariate analysis also showed that these variables were significantly different between the mild/moderate severe group and the severe group, but did not pass the LASSO regression and logistic regression screening. It is possible that these variables contribute less significantly to prediction, or that the sample size is small, or there is case bias.

LASSO regression, a regularization technique, effectively screens variables and simplifies models by imposing a shrinkage penalty term, thereby mitigating overfitting issues arising from multicollinearity (*Zhou, Li & Zhang, 2021*). It is highly efficient and is particularly suitable for high-dimensional data processing (*Emmert-Streib & Dehmer, 2019*). In this study, univariate analysis, LASSO regression, and logistic regression were combined to select three variables from 107 variables, with good model performance, which can help clinicians judge the severity of HLAP. The AUC value of the model built in this study is smaller than that of some studies, but the variables in this study are all commonly

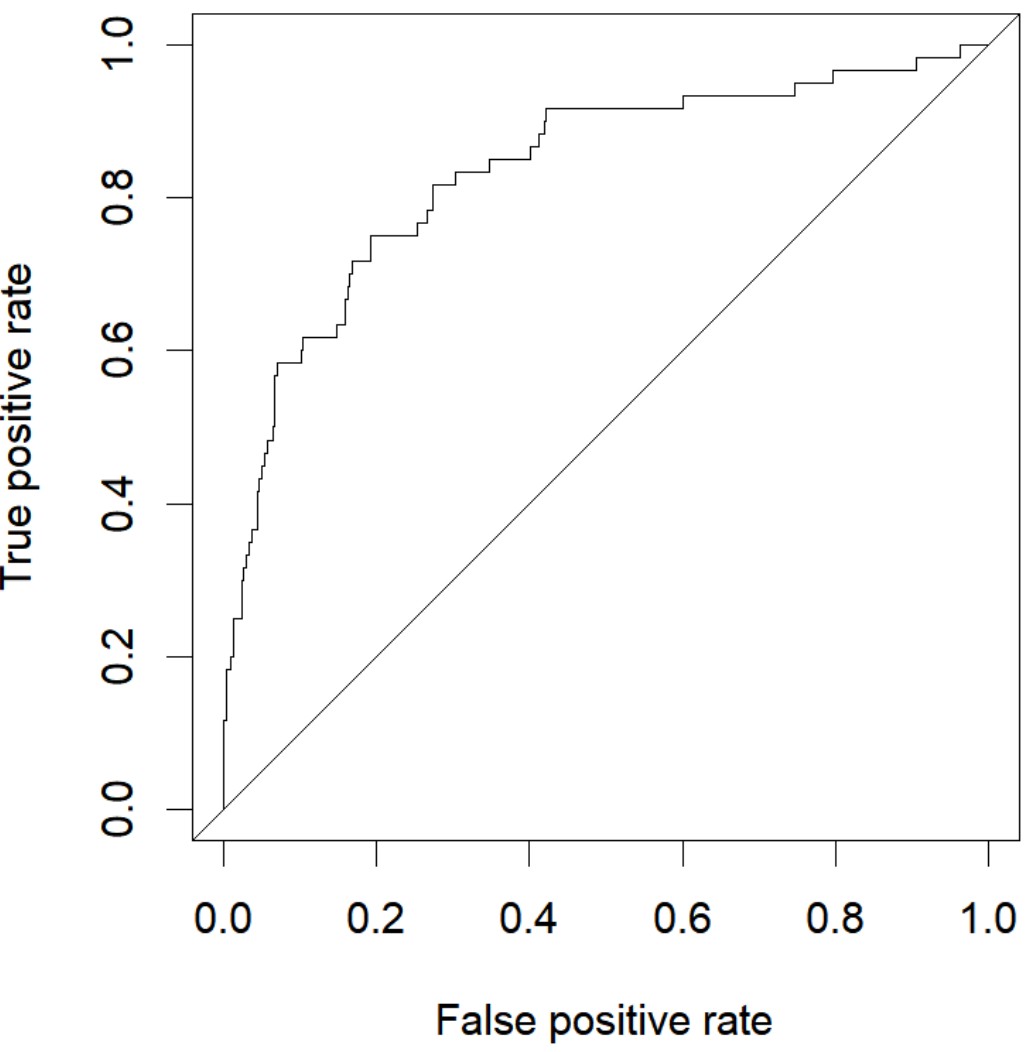

**Figure 3  ROC curve of logistic regression model.**

used clinical indicators, do not include various complex scoring systems, and have the advantages of simplicity and practicality. Moreover, the model does not include pleural effusion, peritoneal effusion pancreatic pseudocyst and other variables occured in the middle and late stages. Therefore, it can be used for early diagnosis.

There are limitations to this study. First, we were unable to collect data on some potential risk factors, such as body mass index, previous surgeries, gene mutations, pain score, narcotic usage, alcohol or smoking habits, endocrine functions, and abdominal fat content and distribution. Second, in this study, a relatively large dataset was collected based merely on empirical judgment, without quantitative approaches (such as estimating learning curves) to assess the sample size. Finally, while our model demonstrated good performance, there is still room for improvement.

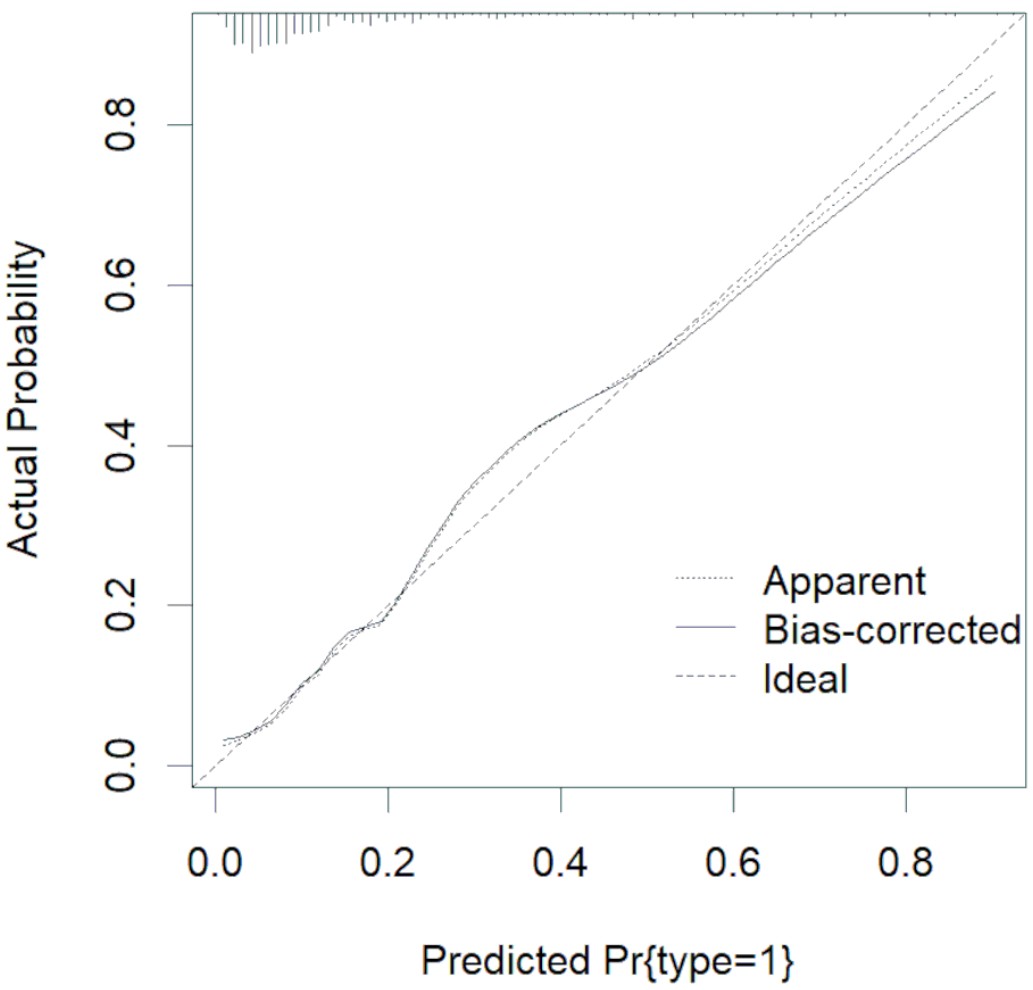

**Figure 4  Calibration curve of logistic regression.**

## CONCLUSIONS

In this study, risk factors for severe HLAP were identified, and a logistic regression model was developed based on LASSO regression. The results showed that the risk factors for severe cases of HLAP included increased D-dimer, decreased serum calcium, and increased cholesterol. The logistic regression prediction model demonstrated robust performance, with an AUC of 0.8341 (95% CI [0.7724–0.8958]), indicating good discriminatory ability. The model outperformed several existing models in terms of simplicity and practicality, thus demonstrating it to be a valuable tool for early identification and intervention by clinicians. Future research should aim to collect more comprehensive data, including additional risk factors, and conduct multicenter external validation to enhance the model's generalizability and accuracy.

## ACKNOWLEDGEMENTS

We would like to express our sincere thanks to the editors and reviewers for their valuable comments. We thank LetPub for its linguistic assistance during the preparation of this manuscript. The authors acknowledge the use the AI tools DeepSeek, Kimi, and Doubao in this study. The AI tools were used solely for literature search (keyword-based initial screening), language proofing (grammar/spelling checks), and manuscript polishing.

### Funding

This work was supported by the Key project of Scientific Research and Development Foundation of Kangda College of Nanjing Medical University (No. KD2023KYJJ187) and the Social Development (Guiding) project of Taizhou Science and Technology Support Plan in 2024 (No. 13). There was no additional external funding received for this study. The funders had no role in study design, data collection and analysis, decision to publish, or preparation of the manuscript.

### Grant Disclosures

The following grant information was disclosed by the authors:
Scientific Research and Development Foundation of Kangda College of Nanjing Medical University: KD2023KYJJ187.
Taizhou Science and Technology Support Plan in 2024: (No. 13).

### Competing Interests

The authors declare there are no competing interests.

### Author Contributions

- Qingyu Zhang conceived and designed the experiments, performed the experiments, analyzed the data, authored or reviewed drafts of the article, and approved the final draft.
- Runping Han analyzed the data, prepared figures and/or tables, data collection and organization, and approved the final draft.
- Zhenxing Li analyzed the data, prepared figures and/or tables, data collection and organization, and approved the final draft.
- Kunfeng Yan analyzed the data, prepared figures and/or tables, data collection and organization, and approved the final draft.
- Xiaorong Dai analyzed the data, authored or reviewed drafts of the article, reviewd the manuscript, and approved the final draft.
- Gongchao Yu conceived and designed the experiments, performed the experiments, analyzed the data, authored or reviewed drafts of the article, reviewd and Revised the manuscript critically, and approved the final draft.

## Human Ethics

The following information was supplied relating to ethical approvals (i.e., approving body and any reference numbers):

This study was approved by the Medical Ethics Committee of Taixing People's Hospital (No: LS2025008; No: YJ2025009) and informed consent was waived.

## Data Availability

Data of HLAP treated in our hospital from January 1, 2020 to June 30, 2023 and the code are available in the Supplemental Files.

## Supplemental Information

Supplemental information for this article can be found online at http://dx.doi.org/10.7717/peerj.20471#supplemental-information.

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
