# Peer review of "Development of a predictive model for severe hyperlipidemic acute pancreatitis based on LASSO regression: a retrospective study"

_PeerJ, doi:10.7717/peerj.20471_

## Round 0.1 · original submission · Major Revisions

· Academic Editor

Major Revisions

**Language Note:** When preparing your next revision, please ensure that your manuscript is reviewed either by a colleague who is proficient in English and familiar with the subject matter, or by a professional editing service. PeerJ offers language editing services; if you are interested, you may contact us at [email protected] for pricing details. Kindly include your manuscript number and title in your inquiry. – PeerJ Staff

Reviewer 1 ·

Basic reporting

The paper is interesting about a timely problem.

Here are some comments to improve the paper:

- Data of HLAP treated in our hospital from: "our hospital" should be replaced by the name of the hospital. Similar expressions should be changed throughout the paper.

- The introduction is very short. More context information and relevant papers should be discussed.

- Was multiple testing correction, e.g., Bonferroni, used? Please discuss this in detail (you test more than one hypothesis).

- Sample size is an important factor for every statistical analysis. So far, this issue has not been addressed in the paper. As a practical approach, learning curves can be used,

I suggest estimating learning curves to investigate the influence of the sample size on the results.

Experimental design

-

Validity of the findings

-

·

Basic reporting

Overall, this study has detailed data, standardized statistical methods, and reliable conclusions. The predictive model constructed in this study has significant advantages in variable selection: all the included indicators are routine variables that are easy to obtain in clinical practice, highlighting its operational simplicity. At the same time, the model does not include mid-to-late-stage characteristic variables such as pleural and ascitic fluid, giving it high application value in the early-diagnosis scenario and strong practicality. The overall framework of the paper is reasonable, but there are still several details that need to be further improved. It is recommended to accept the paper after modification.

The language expression in some parts of the paper needs to be polished. You can contact a professional language polishing team to handle it.

Experimental design

In the research methods, the author mentioned, "The makeX function in the glmnet package of R 4.3.1 software was used to complete the missing values when Lasso regression was performed, and the simple imputation method was used to handle the missing values when a Logistic regression model was constructed." This statement is not detailed enough. It is recommended to supplement:

1. Clarify the specific imputation logic of the makeX function (such as based on the mean, median, or other algorithms);

2. Specify the specific type of the "simple imputation method" (such as mean imputation, mode imputation, or median imputation). This is crucial for evaluating the rationality of the method and the reproducibility of the results.

Validity of the findings

In the conclusion, the expression "a logistic regression model was developed using Lasso regression" is ambiguous. It is recommended to modify it to "a logistic regression model was developed based on Lasso regression". After modification, it can more accurately reflect the logical relationship of "constructing a Logistic model after screening variables using Lasso regression", avoiding the misinterpretation that a Logistic model was directly constructed using Lasso regression.

·

Basic reporting

-

Experimental design

-

Validity of the findings

-

Additional comments

This article is well-designed.
The data is detailed, the results are reliable, and the writing is standardized. However, in the preface, the basis for presenting this research is not sufficient.
The method used lacks novelty.
The final result, when compared with the published literature, lacks innovation.

·

Basic reporting

-

Experimental design

-

Validity of the findings

-

Additional comments

The manuscript is organized well. Provide patients' demographic information. Rearrange Table 1 for more clarity.

Line no - Review comment:

50 - Try to provide the causes of hyperlipidemia (Such as diet habits or genetics, or any other reason)

63 - Provide the details of the Duration of AP for the patient, previous surgeries (ERCP, etc). Gene mutations (CFTR, etc), Pain score, narcotic usage, alcohol or smoking habits, and endocrine functions.

- Provide the kits and machinery details used for diagnosis

---

## Round 0.2 · Minor Revisions

· Academic Editor

Minor Revisions

The authors should include estimating learning curves to further support their statistical findings and small sample size, as previously suggested by Reviewer 1.

I suggest to move the last paragraph in the "Discussion" about limitations and future work, to the "Conclusions". If the authors are not willing to include estimating learning curves as suggested by Reviewer 1, perhaps they can include this as a limitation of their study.

Authors should also be more specific on other limitations here eg. that they failed to collect the following information: previous surgeries (ERCP, etc), Gene mutations (CFTR, etc), Pain score, narcotic usage, alcohol or smoking habits, and endocrine functions. This should also be stated in the limitations, in addition to the potential risk factors already listed (e.g BMI).

·

Basic reporting

no comment

Experimental design

no comment

Validity of the findings

no comment

·

Basic reporting

No comments

Experimental design

No comments

Validity of the findings

No comments

Additional comments

No comments

---

## Round 0.3 · accepted · Accept

· Academic Editor

Accept

The authors have now addressed all the reviewers' suggestions.
This paper is ready for publication.